# Portable Rice Disease Spores Capture and Detection Method Using Diffraction Fingerprints on Microfluidic Chip

**DOI:** 10.3390/mi10050289

**Published:** 2019-04-28

**Authors:** Ning Yang, Chiyuan Chen, Tao Li, Zhuo Li, Lirong Zou, Rongbiao Zhang, Hanping Mao

**Affiliations:** 1School of Electrical and Information Engineering, Jiangsu University, Zhenjiang 212013, China; yangning7410@163.com (N.Y.); chenchiyuan0214@163.com (C.C.); litao0203a@163.com (T.L.); 13037591303@163.com (Z.L.); wxzlr1996@163.com (L.Z.); 2School of Agricultural Equipment Engineering, Jiangsu University, Zhenjiang 212013, China

**Keywords:** crop disease, lensfree, light diffraction, image processing, microfluidic

## Abstract

Crop diseases cause great harm to food security, 90% of these are caused by fungal spores. This paper proposes a crop diseases spore detection method, based on the lensfree diffraction fingerprint and microfluidic chip. The spore diffraction images are obtained by a designed large field of view lensless diffraction detection platform which contains the spore enrichment microfluidic chip and lensless imaging module. By using the microfluidic chip to enrich and isolate spores in advance, the required particles can be captured in the chip enrichment area, and other impurities can be filtered to reduce the interference of impurities on spore detection. The light source emits partially coherent light and irradiates the target to generate diffraction fingerprints, which can be used to distinguish spores and impurities. According to the theoretical analysis, two parameters, Peak to Center ratio (PCR) and Peak to Valley ratio (PVR), are found to quantify these spores. The correlation coefficient between the detection results of rice blast spores by the constructed device and the results of microscopic artificial identification was up to 0.99, and the average error rate of the proposed device was only 5.91%. The size of the device is only 4 cm × 4 cm × 5 cm, and the cost is less than $150, which is one thousandth of the existing equipment. Therefore, it may be widely used as an early detection method for crop disease caused by spores.

## 1. Introduction

Rice is the most important crop all around the world. Currently, approximately 5.6 billion people (80% of the world’s population) have rice as their staple food. Rice consumption is expected to reach 8 million tons in 2030 owing to population and economic growth [1]. The blast fungus causes a serious disease on a wide variety of grasses including rice, wheat, and barley [2]. Rice blast disease, which is caused by a kind of spore named Magnaportheoryzae [3], affects most of the rice-producing countries and has already spread to approximately 85 countries [4]. The severity of rice diseases has been unprecedented in the past 20 years, particularly in China’s Yangtze River basin region in 2018. Most rice disease spores are airborne microparticles with a size of 10–20 μm, and are difficult to detect and capture. Therefore, it is necessary to establish a rapid and effective spore detection facility to prevent crop diseases.

Several methods have been proposed to detect spores which cause rice diseases. Spore analysis is a powerful technique and has been widely used in plant diseases control [5]. In these researches, identification of spores, which can quantitatively evaluate the effect of disease control, is a routine and essential tache in spore assays [6]. Polymerase chain reaction (PCR) is widely used in spore detection due to its high quantitative analysis ability [7]. Rice is a widely cultivated food crop. Although the accuracy of PCR detection is high and the speed is fast enough, the method needs professional personnel and to be sent back to the laboratory for testing, which is not suitable for large-scale promotion in the field. The staining method is also often used in cell and spore identification. The staining agent can impregnate the target and mark it [8], which requires various special stains and destroys the tissue smears. Micromechanical cantilever arrays can be used to detect spore concentration rapidly and quantitatively. This method has the characteristics of high precision and high sensitivity, but it requires a clean detection environment [9]. The common method is to visually measure and count the spores under microscope [10], but the microscope is expensive and bulky, and cannot be widely used outdoors. Most of these methods require complex procedures and multiple reagents, resulting in low experimental repeatability [11]. Also, these complex experimental procedures are time-wasting and labor-intensive [12].

To improve the above problems, developing a convenient and intelligent low-cost spore detection facility has become a hotspot of the filed. Lensless imaging technology is valuable for spore research because in these research fields the statistical regularities are based on the behavior of large populations. Therefore, it is concerned by researchers and applied in the fields of blood cell counting [13] and viability detection [14], particle size classification [15], biofilm detection [16], etc. Traditional optical microscopy has a complex lens structure, so it is not suitable for large-scale popularization because of its large volume and high price. Its precise optical part is sensitive to external conditions such as temperature and humidity, and it is difficult to adapt to complex field environments. Compared with the microscopic imaging mode, lensless imaging technology only uses a complementary metal oxide semiconductor (CMOS) module, which is small in size, stable in performance, and not susceptible to environmental impact. Traditional microscopic imaging has a very small field of view (FOV), usually less than 0.5 mm^2^. Without moving the lens, it is difficult to cover the whole enrichment area of microfluidic chip. The active imaging area of the CMOS sensor is about 20 mm^2^, which means the FOV is about 20 mm^2^ and it is approximately 100 times larger than that of a conventional optical microscope. Different to the field of biomedicine, there are many impurities in the air. If the spores are collected directly without filtering, a large number of impurities will be mixed into the samples, which will reduce the accuracy of image detection in the later stage. Therefore, the microfluidic chip must be added to the spore collection process to separate most impurities. Microfluidic chip has many advantages, such as little reagent consumption, fast automatic detection, low cost, high integration, and has allowed researchers concentrate the whole process of a biochemical reaction on the chip [17]—mostly applied in the field of liquid and medical testing [18]. Because PM2.5, PM10, and other air pollutants have a great impact on human health, researchers have designed a variety of microfluidic chips for the filtration and enrichment of airborne microparticles [19,20]. However, few microfluidic chips have been developed to enrich spores of crop diseases. In this paper, a microfluidic chip is designed to enrich and separate airborne spores. According to the above technology, a spore detection device has been developed, which includes the microfluidic chip and lensless imaging module. It can enrich and detect spores and realize early monitoring of crop diseases.

## 2. Materials and Methods

### 2.1. Design of Microfluidic Chip

The microfluidic device is fabricated by the standard soft-lithography technique [21,22]. The master mold is created on a silicon substrate mold by exposing the photoresist SU-8. After being peeled off from the mold, the polydimethylsiloxane (PDMS) slab is punched through to make ports at the inlet and outlet. The plastic tubes, with a small amount of glue at their ends, are inserted through the inlet and outlet ports. The PDMS slab is treated with oxygen plasma and then bonded to a glass substrate (20 mm × 55 mm). Finally, the assembled device is cured at 70 °C for 30 min to enhance bonding. A sectional view of the enrichment and separation of the microfluidic chip is shown in Figure 1. It has a two-stage separation–enrichment structure to simplify the fabrication process and reduce the costs. L1, L2, L3, and L4 represent the width of channel 2, channel 3, channel 4, and channel 5, respectively. The value of D1 is collection tank 1 entrance width minus channel 2 width. The value of D2 is width of collection tank 2 entrance minus width of channel 4. The arc channel is composed of two concentric arcs with different radii. R1 and R2 are the radius of the inner arc of the arc channels respectively. To obtain better experimental results, 16 μm particles were used to represent rice blast spores and 2 μm particles were utilized to represent impurities in the environment. When particles pass through channel 1 and Arc channel 1, the force on the particles is the same because of the same flow rate of the two channels, but the direction of the force was changed due to the change of the shape. The two particles with different sizes deviate from the trajectory due to the force in two different directions, thus realizing the original separation of the two particles. The 16 μm particles with a larger particle size enter collection tank 1 due to their larger velocity. In order to keep the 16 μm particles in collection tank 1 and 2 μm particles into the second structure, L1 should be smaller than L2. The shape and design principle of the first and second structure are the same, so L3 should be smaller than L4.

### 2.2. Numerical Method

The commercial COMSOL Multiphysics 5.1 was utilized to simulate the flow field and particle motion of microfluidic chip. For numerical analysis, the model parameters were set at width of particle entrance = 2000 μm, length of channel 1 = 12000 μm, R1 = 1000 μm, width of Arc channel 1 = 2000 μm, length of channel 3 = 8000 μm, R2 = 1050 μm, width of Arc channel 2 = L2, length of channel 4 = 10000 μm, length of channel 5 = 5000 μm. All microchannels are 100 μm in height. The parametric study was performed by varying the L2/L1, L4/L3 ratio in the range of 1–2 and D1, D2 value in the range of 800–1800 μm to determine the effect of enrichment and separation.

The flow was assumed to be steady, two-dimensional axisymmetric, and incompressible. Moreover, Reynolds numbers of collection tank 1 and 2 were 173 and 346, respectively. The temperature was 293.15 K and the applied pressure was 101.3 kPa. The velocity inlet condition at the particle entrance, pressure outlet condition at the particle exit, and no-slip condition in all walls were set as the boundary conditions. The laminar inflow rate is 115 mL/min. Continuity, momentum, and energy equations were solved repeatedly by adjusting the convergence criterion at 10^−6^.

After flow analysis, the behavior of particles was investigated. For the particle behavior analysis, the COMSOL Multiphysics Particle Tracing Module was used. A particle was assumed to be permanently collected when it hit any wall. The particle density was set at 1000 kg/m^3^ to express the aerodynamic size. Spore concentration in the air is very low, so 40 particles (2 μm particles and 16 μm particles, 20 of each) were arranged at regular intervals from the center of the particle entrance to the edge. The Lagrangian method is convenient for describing particle movement. The single particle motion equation is expressed as follows:(1)$$m_{p}\frac{d^{2}r_{p}}{dt^{2}}=F_{f}+F_{o}.$$

In the formula, ***r_p_*** is the particle motion position vector, *m_p_* is the particle mass, and *t* is the particle motion time. *F_f_* is the force of fluid on particles and *F_o_* is the force of an external potential field on particles. The particle motion can be followed by solving the particle motion Equation (1).

### 2.3. Diffraction Imaging Detection Platform Setup

The detection facility designed in this paper has the characteristics of small size, low-cost, and convenience. The schematic diagram is shown in Figure 2. The facility has main two parts—the microfluidic chip and the lensless imaging module. The nano-microspheres (1–4 μm), pollen, dust, and spore-carrying rice grains were mixed with water and put into the aerosol generator to produce aerosol. Then, the aerosol was passed through the drying tube to eliminate the water vapor. Then, the dried mixture gas was passed through the flowmeter to obtain stable gas flow rate. At this stage the mixed gas finally enters the microfluidic chip.

The spores were separated and enriched by external force through the microfluidic chip, shown in Figure 2. The microfluidic chip consists of two red-labeled enrichment regions and three microchannels with different diameters. The lensless imaging module includes the light source and CMOS sensor. OSRAM’s LA E65B LED (617 nm) is selected as the light source. The diameter of the micropore was 100 μm directly below it. The CMOS sensor was located 45 mm below the micropore. The MT9P031 image sensor chip with 5 million pixels from Aptina was selected. The size of the imaging area was 5.70 mm × 4.28 mm, which means the FOV was up to 24 mm^2^ and approximately 100 times larger than that of a 200× conventional microscope. The LED light source was monochrome, so a black-and-white version of the chip was customized to obtain gray image directly to improve the signal-to-noise ratio of the image. In addition, a drawer-type sample tray was designed to realize quick replacement of the microfluidic chip. When using the device, the air pump was turned on to collect the spore mixture gas into the microfluidic chip for enrichment and separation. Then, the lensless imaging module photographed the diffraction fingerprints of enriched spores in the microfluidic chip and transferred the image to PC for image processing and saving. The facility realizes the independent operation of spore enrichment, sampling, photographing, uploading, and detection.

### 2.4. Spore Detection Using Diffraction Parameters

In order to design a spore detection system based on spore diffraction fingerprint, the acquisition and image processing of the diffraction fingerprint were automatic. The steps of acquisition and analysis of lensless diffraction fingerprint information are as follows: (1) Spores are enriched by microfluidic chip; (2) the enrichment area of microfluidic chip is placed on the CMOS sensor; (3) high definition diffraction fingerprint images are acquired by CMOS sensor; (4) diffraction fingerprint images are transferred to computer for further processing as described in the following steps: a. Images are read sequentially; b. the diffraction fingerprints of each spore are extracted from the picture; c. characteristic parameters are calculated by using pixel information of diffraction fingerprint; d. comparing the calculated characteristic parameters with the threshold value of spore characterization to determine whether the target is the spore or not. There is no need to add additional reagents and the whole process is time-saving and labor-saving.

In the study (shown in Figure 3), two parameters, Peak to Center ratio (PCR) and Peak to Valley ratio (PVR), were designed to quantify these spore fingerprint characteristics and are formulized as follows, respectively:(2)$$PCR=\frac{AP}{C},$$
(3)$$PVR=\frac{AP}{AV}.$$

AP is the average relative light intensity of the two peaks (P1, P2) in a single spore fraction fingerprint. AV is the average relative light intensity of the two lowest points (V1, V2). The dark fringe nearest to the center (C) is called the main dark fringe, and the bright fringe is called the main bright fringe.

The parameters were employed to distinguish spores and other impurities. Matlab was utilized to extract spore fingerprints and calculate the parameters corresponding to each spore. A spore was identified as a rice blast spore only when the two calculated parameters satisfy both of the two conditions that PCR and PVR were between the determined threshold. The threshold levels of every specific spore lines should be determined by comparing the average PCRs and PVRs of rice blast spores in advance. In the study, the two threshold levels of the selected rice blast spores’ lines were determined to be 1.25–1.40 of PCR and 3.35–3.5 of PVR from 2000 spores of rice blast. The spore, whose corresponding fingerprint parameters satisfy the thresholds given above, could be identified as a rice blast spore.

## 3. Results and Discussion

### 3.1. Simulation and Experiment of Microfluidic Chip

#### 3.1.1. Particle Motion Simulation

Figure 4a,b shows the relationship between the separation and enrichment effect of 16 μm particles and L1, and the relationship between the separation and enrichment effect of 16 μm particles and L2/L1. The D1 is 500 μm, the L1 and value of L2/L1 is changed. The separation and enrichment effect of 16 μm particles are better when L1 = 1500, 1400, and 1300 μm; and the separation and enrichment effect of 16 μm particles are best when L2/L1 = 1.4.

In order to study the relationship between the separation and enrichment effect of 2 μm particles and L3, the relationship between the separation and enrichment effect of 2 μm particles and L4/L3 must also be studied. Figure 4c only shows L3 = 700, 800, 900, 1000 μm because when L3 takes other larger values, 2 μm particles cannot be collected, but all reach the exit. When L3 = 700 μm, the separation and enrichment effect of 2 μm particles are better regardless of the value of L4/L3; when L3 takes different values, there is no uniform L4/L3 value that can make the best separation and enrichment effect under these conditions.

In order to study the effect of channel 2 length on particle enrichment and separation, a variable channel 2 length (2000–10000 μm) was set up. As shown in Figure 4d, when the length of channel 2 is 5000 μm, 6000 μm, and 7000 μm, most 16 μm particles can enter the 16 μm collection tank, and most 2 μm particles can enter the next structure. When channel 2 is too short, all particles cannot obtain enough centrifugal force, so it is difficult for particles to enter the collection area. When channel 2 is too long, the centrifugal force obtained by the particles will be too large, and the small particles will also enter the 16 μm collection area. Finally, the length of channel 2 is 5000 μm.

In order to study the relationship between the separation and enrichment effect of 16 μm particles and the D1, the L2/L1 is 1.4. By changing the value of D1, the range of D1 is 100–1000 μm. When L1 changes from 1900 μm to 1000 μm, the number of 16 μm particles collected is studied. Figure 5a,b shows that the D1 of optimum separation and enrichment is different when L1 is different, and 200, 300, and 500 μm can be regarded as the optimum D1.

Figure 5c shows the relationship between the separation and enrichment effect of 2 μm particles and D2. The number of 2 μm particles collected under different L3 conditions was studied. The best separation and enrichment effect are obtained at D2 = 1000 μm.

In order to determine the final structure of microfluidic chip, the optimal structure parameters should be further selected. For the primary structure of collecting 16 μm particles, the optimum L2/L1 is 1.4. Considering that the small size of micro-structure will increase the complexity of the fabrication process, L1 = 1500 μm is chosen as the optimum channel width and channel 2 is 5000 μm in length. When L1 = 1500 μm, D1 has little effect on the separation and enrichment of particles. In order to simplify the structure, the D1 is 500 μm. For the secondary structure of collecting 2 μm particles, only when L3 = 700 μm can the separation and enrichment effect be better. When L3 = 700 μm, L4/L3 has little effect on the separation and enrichment of particles. In order to simplify the structure, L4/L3 = 1.4 is chosen. The optimum D2 is 1000 μm. Figure 6a is the simulation result of the separation and enrichment effect of two kinds of particles, 20 particles each. 19 particles of 16 μm were collected in collection tank 1, and 19 particles of 2 μm was collected in collection tank 2. Figure 6b is the velocity distribution in the microchannel of the chip.

As shown in Figure 6, the principle of separation and enrichment of the first structure is first analyzed. The mixed gas enters the microchannel from the inlet of the microfluidic chip and obtains a horizontal to right initial velocity. If there is no Arc channel 1, the particles cannot get enough centrifugal force to enter collection tank 1 at a right angle turn. When the Arc channel 1 is added, the Arc channel 1 has a coupling effect with the right angle turn, which increases the centrifugal force of the particles, so that it can enter collection tank 1. In the simulation, the density of all the particles is the same, so the larger the diameter of the particles, the greater the centrifugal force. The 16 μm particles can enter collection tank 1 due to their large centrifugal force, while the 2 μm particles can enter the second structure. Then the principle of separation and enrichment of the second structure is analyzed. Similar to the first structure, Arc channel 2 and the right angle turn produce a coupling effect, which increases the centrifugal force of the particles and facilitates entry into collection tank 2. Reducing L3 accelerates the particles, increasing the acceleration of the 2 μm particles and increasing the centrifugal force to enter collection tank 2.

#### 3.1.2. Spore Collection Experiment

Figure 7a,b is an experimental image taken by a scanning electron microscope. Rice blast spores are collected in collection tank 1. The target in the red circle is the rice blast spore. The spore samples of rice blast were cultured from rice grains, so the fine impurities in collection tank 1 were rice husks, as shown in the black circle marker (not all markers for the clarity of the picture), which affect the shooting effect. Other impurities are collected in collection tank 2. Therefore, the designed microfluidic chip can separate and collect rice blast spores.

### 3.2. Diffraction Imaging Detection Platform

#### 3.2.1. Diffraction Fingerprints Calculation and Investigation

Traditional optical microscopy imaging technology is based on the change of wavelength (color) and amplitude (brightness) when light passes through transparent substances. The morphology of microorganisms is directly observed by the naked eye through microscopy. The light emitted by LED is converted into coherent light through micro-holes, thus realizing coherent illumination. As shown in Figure 2, the light emitted by the LED light source passes through the micropore directly below it, and produces partially coherent light. The partially coherent light propagates over a certain distance and then irradiates on the sample plane (microfluidic chip). Diffraction hologram is formed by the interference of the sample’s backlight and scattered light on the sample plane, which is photographed by CMOS imaging chip directly below the sample plane.

The classical scalar diffraction theory was first proposed by Huygens in 1678. Huygens believed that any point on the wave surface could be regarded as the source of secondary spherical wavelet in the process of light propagation. Fresnel further proposed in 1818 that the interference between the wavelet source and the wavelet source should be superimposed. Hence, the famous Huygens–Fresnel principle is proposed that the light vibration at any point outside the wavefront should be the result of the coherent superposition of all wavelets on the wavefront. The Huygens–Fresnel principle can be utilized to explain the diffraction of light. Its mathematical expression can be written as follows:(4)$$U\left(Q\right)=c\iint_{\sum }U_{0}\left(P\right)K\left(\theta \right)\frac{e^{jkr}}{r}dS.$$

Among them, *S* is the surface of object wavefront. In this work, the object wavefront *S* is formed by light source transmitting through spore sample. *c* is a constant, *∑* is an integral plane, *U*_0_(*P*) is the complex amplitude of *P* at any point on the wavefront, *U*(*Q*) is the complex amplitude of *Q* at any point in the light wave field, *r* is the distance from *P* to *Q*, *θ* is the angle formed by the normal *N* and *PQ* at the wavelet front where P is located, and *K* (*θ*) is the tilt correlation between *P* and *Q*.

MATLAB was utilized to calculate the diffraction fingerprint with the same parameters setting as that of the experimental setup. Fourier transform and discretization are used and suitable sampling intervals are selected in calculation. Figure 8a shows that the spore image is processed into an impermeable boundary and analyzed as the pattern of a diffraction screen. Firstly, the contour of the spore is displayed by setting the gray threshold through the grayscale display, then the noise in the image is removed by setting the threshold of the boundary pixel point, and the blank vacancy in the spore is filled. Finally, a similar spore pattern is generated through the black and white inversion. The generated spore contour image is calculated using the formula (4) to obtain the image shown in Figure 8b. The ordinate represents the relative intensity of the light field. It can be seen that the amplitude of the central light field is high, the amplitude of the edge is low, and the periphery of the edge presents a state of high and low alternation. In other words, in the center of the captured pattern the diffraction fringes are distinct, while at the edge the fringes are effectively eliminated. The reason is that the partially coherent light source contains rays of different phase. The rays lead to different fringe distribution at the edge of the pattern. The result is good for diffraction fingerprint quality because it reduces spore’s signatures overlap. The center area and these main fringes are more valuable.

A rice blast spore can be approximately regarded as an oval. The observations indicate that spore morphology has a close relationship with the diffraction fingerprint. The spore models were set to ellipses, which had the same coverage area but different short and long axes. The ratio of semi-minor axis length is 1:1, 1.5:1, 2:1, 2.5:1, and 3:1. Their diffraction intensity patterns are shown in Figure 9. As the ratio of semi-minor axis length increases, the position of the main dark fringe and the main bright fringe changes little, while the brightness of the main dark fringe of the diffraction fingerprint decreases and the brightness of the main bright fringe increases. The above changes will cause changes in PCR and PVR parameters.

According to the analysis above, two parameters, PCR and PVR are designed to quantify the Peak to Center ratio and Peak to Valley ratio. The two computational expressions are described in formulas (1) and (2).

#### 3.2.2. Validation and Superiority of Proposed Approach

Gas mixed rice blast spores were prepared by using an aerosol generator. Then, mixed gas was pumped into the designed microfluidic chip and the performance of the device was verified by taking pictures. Collection tank 2 of the microfluidic chip was observed under a microscope. Figure 10a is a spore sample which has not been enriched and separated by microfluidic chip. There are more small particle impurities in it than in Figure 10b. Excessive impurities will cause adverse effects such as overlap, as Figure 10c shows, which will affect the accuracy of subsequent diffraction imaging detection of spores. The detection accuracy of samples without microfluidic chip enrichment is about 80%, and that of samples with microfluidic chip enrichment is 94%. Because of the high concentration of spores in the disposed spore solution and less other impurities, better results can be obtained without using the microfluidic chip for enrichment and separation. However, in the field environment there are many impurities, low spore concentration and excessive impurities will make the detection accuracy less than 50%. So, it is necessary to use microfluidic chip to pre-filter samples before detection. To assess the validity and reliability of a screening method which used the two parameters as a criterion for spore screening, 20 experimental samples mixed with impurities such as pollen, dust, and beads were selected in the assay. The two parameters, PCR and PVR, were computed from the corresponding diffraction fingerprints. Twenty groups of experiments were conducted. Figure 10d is the result of the test. The manual counting results and automatic counting results are shown in Figure 10e. It can be concluded that the error rate ((Automatic counting result–Manual counting result)/ Manual counting result) fluctuates around 5%, with a maximum of 10% and an average error of 5.91%, which is an acceptable range. The correlation coefficient (R^2^) of the 20 groups of test results is 0.9912, and the automatic counting method is highly correlated with the manual counting method. Figure 10f is a cluster analysis of rice blast spores and 2 μm beads using PVR and PCR parameters. It can be concluded that the two substances are clearly classified. Figure 10g is the result of Bland–Altman analysis of the two methods. All the points are within the 95% consistency interval. These analysis results reveal a good agreement between the two modalities and prove the validity of this approach. Furthermore, the proposed approach is a continuous detection method, and the cost of our constructed platform is far lower than that of other techniques.

## 4. Conclusions

In summary, an approach which can enrich and distinguish spores utilizing a microfluidic chip and lensfree spore diffraction fingerprints in an automatic manner is demonstrated. The designed microfluidic chip can enrich rice blast spores. The two parameters, PCR and PVR, were firstly designed to quantify the characteristics of spore diffraction fingerprint and validated to screen spores. Both of the proposed approaches and manual counting methods were employed to measure spore. The high correlation of 0.9912 and Bland–Altman analysis revealed a good agreement and validated the proposed approach. The detection accuracy of rice blast spores was 94%. The cost of our constructed platform is far lower than others, and no extra operation is required during the detection process. This approach is high-automatic, continuous, time-saving, and labor-saving. These results and advantages illustrate that the proposed approach may be widely used for fungal disease detection caused by spores.

## Figures and Tables

**Figure 1 micromachines-10-00289-f001:**
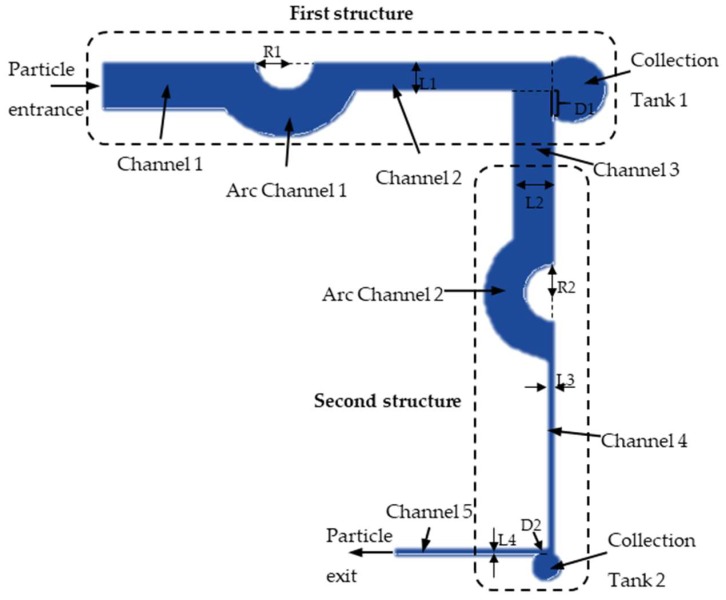
Structural chart of microfluidic chip.

**Figure 2 micromachines-10-00289-f002:**
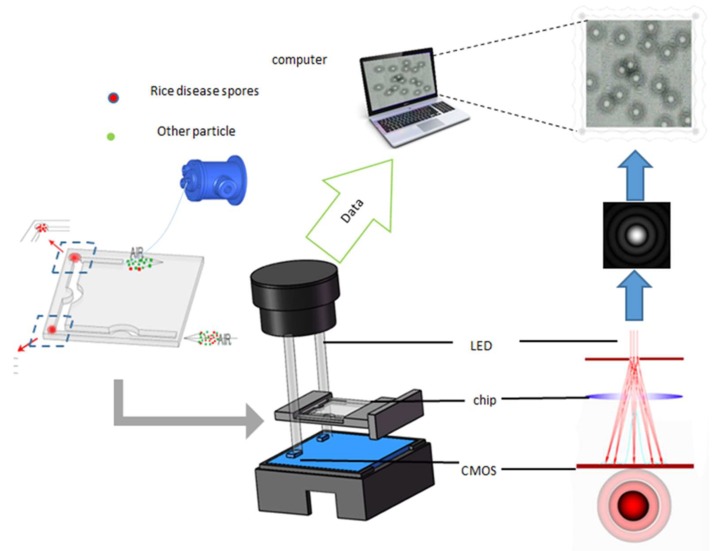
Portable real-time rice disease spores detection using characteristics of diffraction fingerprints on the microfluidic chip.

**Figure 3 micromachines-10-00289-f003:**
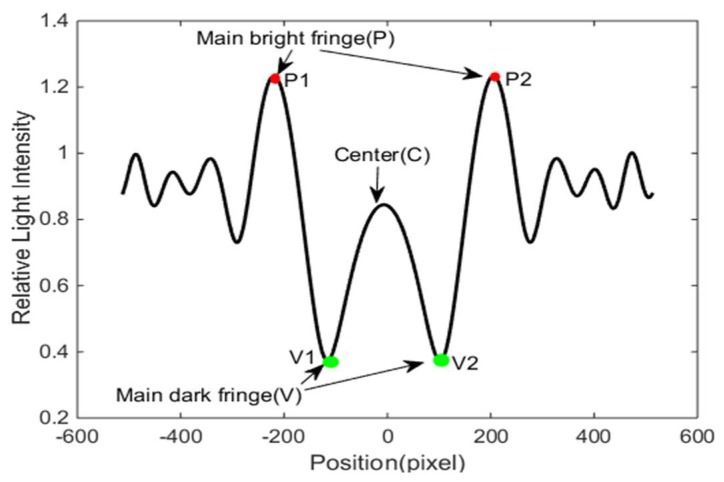
The light intensity value profile of calculated spore diffraction fingerprint.

**Figure 4 micromachines-10-00289-f004:**
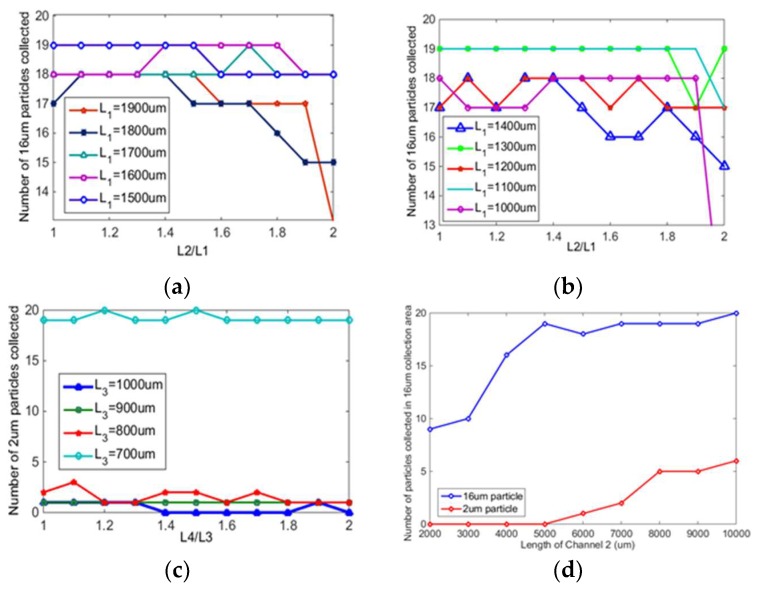
(**a**,**b**) Separation and enrichment of 16 μm particles; (**c**) separation and enrichment of 2 μm particles; (**d**) effect of channel 2 length on particle separation and enrichment.

**Figure 5 micromachines-10-00289-f005:**
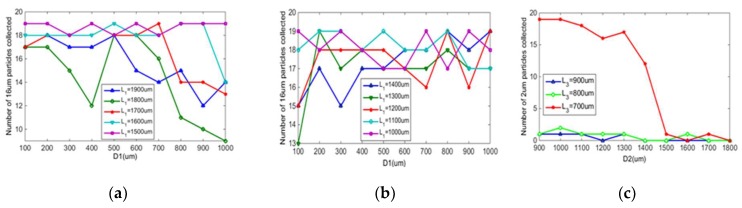
(**a**,**b**) Relationship between the separation and enrichment effect of 16 μm particles and D1; (**c**) relationship between the separation and enrichment effect of 2 μm particles and D2.

**Figure 6 micromachines-10-00289-f006:**
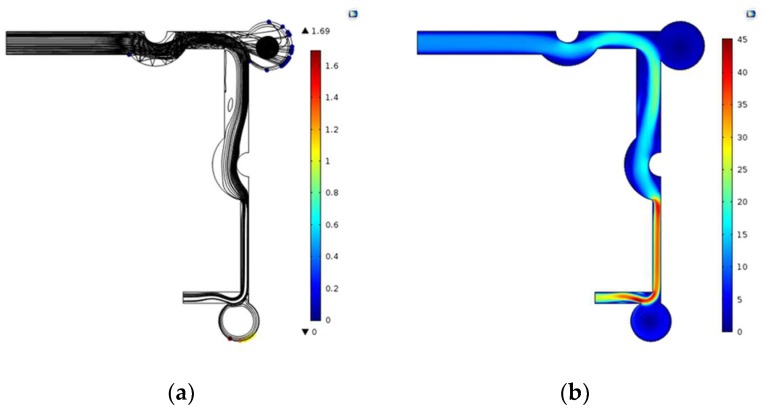
(**a**) Particle trajectory simulation in microfluidic chip; (**b**) simulation of velocity distribution in microfluidic chip (m/s).

**Figure 7 micromachines-10-00289-f007:**
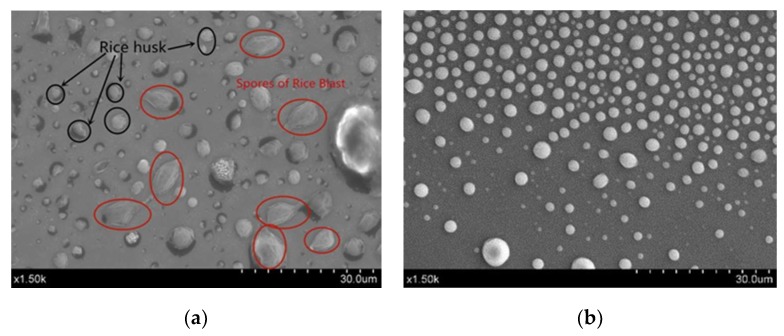
(**a**) Samples collected in collection tank 1; (**b**) samples collected in collection tank 2.

**Figure 8 micromachines-10-00289-f008:**
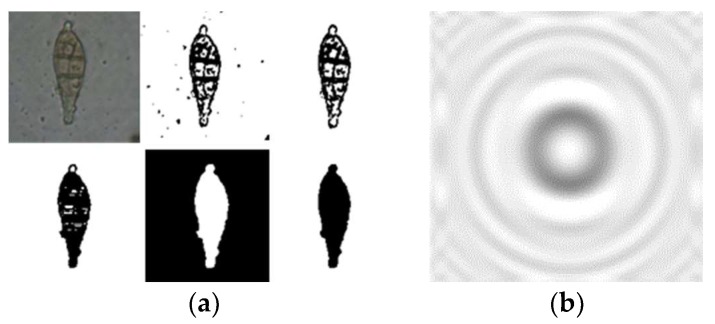
Spore image processing and spore fingerprint captured in this work. (**a**) Gray processing of spore image; (**b**) spore diffraction fingerprint calculation.

**Figure 9 micromachines-10-00289-f009:**
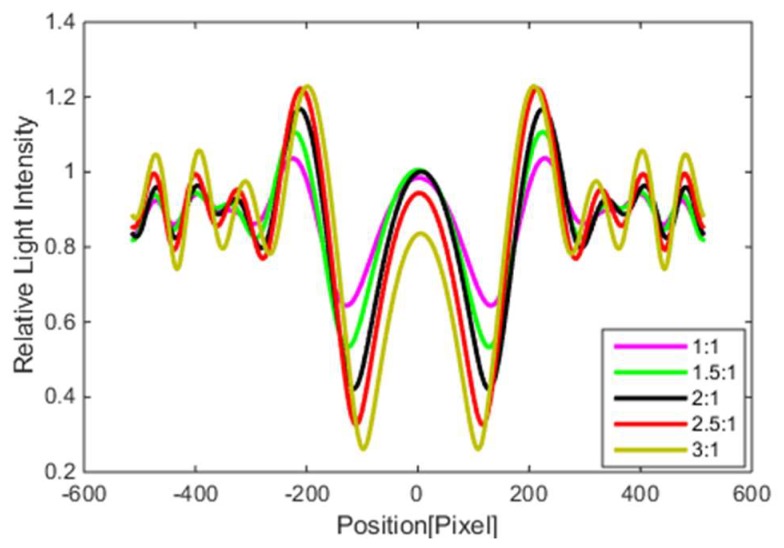
The light intensity value profile of the calculated spore diffraction fingerprint.

**Figure 10 micromachines-10-00289-f010:**
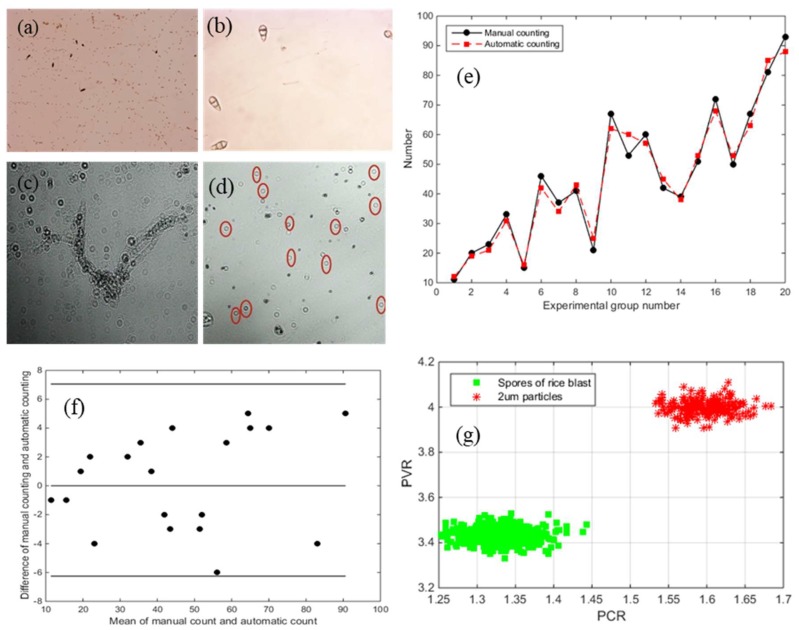
(**a**) Spore samples not enriched and separated by microfluidic chip; (**b**) spore samples enriched and separated by microfluidic chip; (**c**) overlapping interference of diffraction image; (**d**) diffraction image taken by complementary metal oxide semiconductor (CMOS), the fingerprints in red circles correspond to spores; (**e**) experimental result; (**f**) cluster analysis; (**g**) analysis by the Bland–Altman method.

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
