# Peer review of "Portable Rice Disease Spores Capture and Detection Method Using Diffraction Fingerprints on Microfluidic Chip"

_micromachines, 2019, doi:10.3390/mi10050289_

Round 1

Reviewer 1 Report

This paper describes an integrated device for separating and observing the spore particles which cause rice disease. The experimental setup is simple, and the results are validated by simulation and microscopy images.

The paper might be publishable, but some major revisions are needed first to address the following issues:

1.       In section 3.1, are the results from the simulation or experiments? What are the properties of the carrier fluid (line 149) here? In figure 6, what kind of particles are those in the images? Are they real spores, or model particles? All the relevant information is missing in Experimental part, and it is very confusing.

2.       Generally speaking, the authors need to separate the experimental and results/discussion in a clearer way. A lot of experimental details are placed in Section 3, which should be in Section 2. The simulation procedure is also missing in the manuscript.

3.       The principle of the particles separation by the microfluidic channels needs to be clearly explained. What is the resolution of the separation method?

4.       How do the authors define “error size” (line 291)?

5.       How do the authors define “average strength” (line 134)?

6.       The writing of the manuscript is sloppy and needs substantial improvement. Here are a few examples: the spelling of um is wrong; “Peak to valley” should be “Peak to Valley”; the label “Main dark fringe” in figure 3 is partially covered.

Author Response

This paper describes an integrated device for separating and observing the spore particles which cause rice disease. The experimental setup is simple, and the results are validated by simulation and microscopy images.

The paper might be publishable, but some major revisions are needed first to address the following issues:

1. In section 3.1, are the results from the simulation or experiments? What are the properties of the carrier fluid (line 149) here? In figure 6, what kind of particles are those in the images? Are they real spores, or model particles? All the relevant information is missing in Experimental part, and it is very confusing.

Answer: In section 3.1, the experimental results and simulation results have been described in two parts. Spores are first mixed with water and put into the aerosol generator, which produces the gas mixture of spores and enters the microfluidic chip. It does not mean that spores need to be mixed with the carrier fluid to enter the microfluidic chip. Amendments have been made in the text to make the expression clearer. The original experiment used model particles. Later, this part of the experiment was re-done using the spore samples of rice blast, and this part of the experiment was updated. Relevant information has been supplemented in the article.

2. Generally speaking, the authors need to separate the experimental and results/discussion in a clearer way. A lot of experimental details are placed in Section 3, which should be in Section 2. The simulation procedure is also missing in the manuscript.

Answer: Experiments and simulations have been written separately. The experimental part has been adjusted to 3.1.2. The design of simulation parameters and the simulation process are written in 2.2 and 3.1.1.

3. The principle of the particles separation by the microfluidic channels needs to be clearly explained. What is the resolution of the separation method?

Answer: The principle of separation and enrichment of the first structure is first analyzed. The mixed gas enters the microchannel from the inlet of the microfluidic chip and obtains a horizontal to right initial velocity. If there is no Arc channel 1, the particles cannot get enough centrifugal force to enter the collection tank 1 at a right angle turn. When the Arc channel 1 is added, the Arc channel 1 has a coupling effect with the right angle turn, which increases the centrifugal force of the particles, so that it can enter the collection tank 1. In the simulation, the density of all the particles is the same, so the larger the diameter of the particles, the greater the centrifugal force. The 16um particles can enter the collection tank 1 due to their large centrifugal force, while the 2um particles can enter the second structure. Then the principle of separation and enrichment of the second structure is analyzed. Similar to the first structure, the Arc channel 2 and the right angle turn produce a coupling effect, which increases the centrifugal force of the particles and facilitates entry into the collection tank 2. Reducing L3 accelerates the particles, increasing the acceleration of the 2um particles, and increasing the centrifugal force to enter the collection tank 2. Related content has been added in the paper 3.1.1

4.  How do the authors define “error size” (line 291)?

Answer: “error size” is a slip of the pen. The intention is to write the “error rate”. Error rate = (Automatic counting result-Manual counting result)/ Manual counting result. It has been revised and explained in this paper.

5. How do the authors define “average strength” (line 134)?

Answer: “average strength” means “average relative light intensity” is more appropriate. It has been revised and explained in this paper.

6. The writing of the manuscript is sloppy and needs substantial improvement. Here are a few examples: the spelling of um is wrong; “Peak to valley” should be “Peak to Valley”; the label “Main dark fringe” in figure 3 is partially covered.

Answer: “Peak to valley” has been updated to “Peak to Valley”. Figure 3 has been redrawn and updated to this article.

Reviewer 2 Report

This paper presents a microfluidic chip that works in air and that is aimed at the separation of spores from other solid debris and contaminants. This chip is used to detect spores that cause a disease targeting rice. Detection is performed by image analysis using an simple imaging module without lenses. 

The paper in its present form has a very low interest as many key elements are missing:

Lines 37 to 45: The authors explain that other methods exist for analysing spores, but that they are slow and that an analysis takes more than 1h, which seems to be unacceptably long to them.  Could the authors explain why time is so important for the detection of spores in cultures? What are the immediate actions that need to be taken within one hour of the detection of spores to preserve the rice from being infected?

Line 64: The authors use a gas-based microfluidic chip for the separation of spores. There are a number of studies in the field of microfluidics involving the detection of airborne particles. It would be nice if the authors could cite a few of them.

Line 70: The authors never describe the fabrication process they used for their microfluidic chip. This should be done at the beginning of the materials and methods section.

Lines 74 to 92: The dimensions of the fabricated chip should be placed in a table, not in the main text.

Line 83: What is the flow rate  at the inlet, outlet and in the different channels?

Line 83: Does the chip work at only one given value of the flow rate, or for any value of the flow rate at the inlet?

Line 83: What is the Reynolds number in the different parts of the chip?

Line 85: It seems there is a height difference involved here. Is it a two level structure? Where exactly is this difference in the height? Is the bottom part of the channel closed, with the difference in the height creating a step somewhere? Is it a continuous change in height? Is there a junction between a large channel and a small channel?

Lines 70 to 92: What is the principle used in this chip for the particle separation? 

Figure 1: This seems to be the top-view of the chip? Could you also add a view of the flow-velocity profile in the channels, such that we understand the movement of the air in the structure?

lines 104: The 5Mpixel CMOS sensor is certainly the most expensive component in the setup. Why not add a lens? The price of a simple lens would not change much the overall cost of the device, and it would certainly simplify the image processing. In particular image recognition algorithms could be used directly. These algorithms are very powerful in the precise case of this type of image recognition.

Line 149: "spores are dispersed and suspended in the carrier fluid": What exactly does this mean? previously, line64, it was mentioned that this paper was about gas-microfluidics. But now the spores are dispersed and suspended in a carrier fluid. Is this carrier fluid air or something else? If the spores have to be dispersed, this means that they are collected prior to being used in the chip? So how are they collected?

Line 150: "N-S equation"- Is this the Navier-Stokes equation?

Line 151: Electroosmotic flow is the flow resulting from the movement of ions in a liquid submitted to an electric potential. It does not apply for air flows, so why take this example? This is confusing.

Line 160: "according to simulation experiments...". What are "simulation experiments"? Is it computer simulations or experimental measurements? If the authors performed simulations, where are they described? What program was used? What are the conditions?

Lines 163 to 200: The text is impossible to read, because there are so many numerical values in each sentence that it does just not make any sense any more. Please explain the phenomena occurring or the graphs without using numerical values in your sentences.

Figure 4 and 5: These graphs are based on measuring the behaviour of 20 particles. This is not much. Please do these experiments with a large enough number of particles (thousands) such that you have valid statistics.

Line 261 to 268: The authors approximate the spores by oval shapes to calculate the fingerprint of the diffraction pattern. Previously, line143, they explain that they measured the diffraction patterns of 2000 spores to select the threshold levels of the fringes in the corresponding diffraction patterns. This just makes no sense at all. If you have measured 2000 diffraction patterns corresponding to actual spores, why not use them directly? There is no need to approximate the spores by oval shapes. 

Figure 276, legend of figure 8: It is not the profile of spore diffraction fingerprint that is shown here but the profile of oval shapes diffraction fingerprint. 

Line 284: The detection accuracy without microfluidic chip is 80%, which is really good. Using the microfluidic chip changes the accuracy to 94%, which is not much more. So this seems to indicate that there is no real need for a microfluidic system at all!

Lines 287 to 290 confronted to figure 9g: The authors measured 20 samples of spores mixed with pollen and dust, and they seem happy with the results which show a correlation of 0.9912. However, they do not show the results of these experiments in figure 9g, which would have been logical, but the result of another experiment where spores were mixed to beads. Why? Do the authors have something to hide?

In conclusion, there are many points that are not clearly explained, and the methodology used by the authors seems often questionable.  

Author Response

This paper presents a microfluidic chip that works in air and that is aimed at the separation of spores from other solid debris and contaminants. This chip is used to detect spores that cause a disease targeting rice. Detection is performed by image analysis using an simple imaging module without lenses.

The paper in its present form has a very low interest as many key elements are missing:

1. Lines 37 to 45: The authors explain that other methods exist for analysing spores, but that they are slow and that an analysis takes more than 1h, which seems to be unacceptably long to them.  Could the authors explain why time is so important for the detection of spores in cultures? What are the immediate actions that need to be taken within one hour of the detection of spores to preserve the rice from being infected?

Answer: The intention is to explain the inconvenience of the original method. Although the accuracy of PCR detection is high and the speed is fast enough, the method needs professional personnel and sent back to the laboratory for testing, which is not suitable for large-scale promotion in the field.

2. Line 64: The authors use a gas-based microfluidic chip for the separation of spores. There are a number of studies in the field of microfluidics involving the detection of airborne particles. It would be nice if the authors could cite a few of them.

Answer: Documents on the detection of air pollutants such as PM2.5 and PM10 by microfluidic chips have been supplemented in this paper.

3. Line 70: The authors never describe the fabrication process they used for their microfluidic chip. This should be done at the beginning of the materials and methods section.

Answer: the fabrication process for the microfluidic chip has been added to the beginning of the materials and methods section. The microfluidic device is fabricated by the standard soft-lithography technique [21, 22]. The master mold is created on a silicon substrate mold by exposing the photoresist SU-8. After being peeled off from the mold, the polydimethylsiloxane (PDMS) slab is punched through to make ports at the inlet and outlet. The plastic tubes with a small amount of glue at their ends are inserted through the inlet and outlet ports. The PDMS slab is treated with oxygen plasma and then bonded to a glass substrate (20 mm × 55 mm). Finally the assembled device is cured at 70 °C for 30 min to enhance bonding.

4. Lines 74 to 92: The dimensions of the fabricated chip should be placed in a table, not in the main text.

Answer: The dimensions of the fabricated chip should be placed at the beginning of 2.2, and a large number of repetitive writing parts have been deleted. It also has strong readability without using tables.

5. Line 83: What is the flow rate at the inlet, outlet and in the different channels?

Answer: The flow rate at the chip inlet is 115 ml/min and remains unchanged.

6. Line 83: Does the chip work at only one given value of the flow rate, or for any value of the flow rate at the inlet?

Answer: The chip work at only one given value of the flow rate which is 115 ml/min.

7. Line 83: What is the Reynolds number in the different parts of the chip?

Answer: Reynolds numbers of collection area 1 and 2 are 173 and 346 respectively.

8. Line 85: It seems there is a height difference involved here. Is it a two level structure? Where exactly is this difference in the height? Is the bottom part of the channel closed, with the difference in the height creating a step somewhere? Is it a continuous change in height? Is there a junction between a large channel and a small channel?

Answer: Here is the misunderstanding caused by my unclear description. The microfluidic chip designed in this paper has only one layer. The name of height difference is not good. Now we use D1 and D2 instead. As shown in Figure 1, the value of D1 is collection tank 1 entrance width minus channel 2 width. The value of D2 is width of collection tank 2 entrance minus width of channel 4.

9. Lines 70 to 92: What is the principle used in this chip for the particle separation?

Answer: The principle of separation and enrichment of the first structure is first analyzed. The mixed gas enters the microchannel from the inlet of the microfluidic chip and obtains a horizontal to right initial velocity. If there is no Arc channel 1, the particles cannot get enough centrifugal force to enter the collection tank 1 at a right angle turn. When the Arc channel 1 is added, the Arc channel 1 has a coupling effect with the right angle turn, which increases the centrifugal force of the particles, so that it can enter the collection tank 1. In the simulation, the density of all the particles is the same, so the larger the diameter of the particles, the greater the centrifugal force. The 16um particles can enter the collection tank 1 due to their large centrifugal force, while the 2um particles can enter the second structure. Then the principle of separation and enrichment of the second structure is analyzed. Similar to the first structure, the Arc channel 2 and the right angle turn produce a coupling effect, which increases the centrifugal force of the particles and facilitates entry into the collection tank 2. Reducing L3 accelerates the particles, increasing the acceleration of the 2um particles, and increasing the centrifugal force to enter the collection tank 2. Related content has been added in the paper 3.1.1

10. Figure 1: This seems to be the top-view of the chip? Could you also add a view of the flow-velocity profile in the channels, such that we understand the movement of the air in the structure?

Answer: As shown in Figure 6(b), a view of the flow-velocity profile in the channels has been added to the text.

11. lines 104: The 5Mpixel CMOS sensor is certainly the most expensive component in the setup. Why not add a lens? The price of a simple lens would not change much the overall cost of the device, and it would certainly simplify the image processing. In particular image recognition algorithms could be used directly. These algorithms are very powerful in the precise case of this type of image recognition.

Answer: Traditional optical microscopy has complex lens structure, so it is not suitable for large-scale popularization because of its large volume and high price. Its precise optical part is sensitive to external conditions such as temperature and humidity, and is difficult to adapt to complex field environment. Compared with the microscopic imaging mode, lensless imaging technology only uses a complementary metal oxide semiconductor (CMOS) module, which is small in size, stable in performance and not susceptible to environmental impact. Traditional microscopic imaging has a very small field of view (FOV), usually less than 0.5 mm2. Without moving the lens, it is difficult to cover the whole enrichment area of microfluidic chip. The active imaging area of the CMOS sensor is about 20 mm2, which means the FOV is about 20 mm2 and it is approximately 100 times larger than that of a conventional optical microscope.

12. Line 149: "spores are dispersed and suspended in the carrier fluid": What exactly does this mean? previously, line64, it was mentioned that this paper was about gas-microfluidics. But now the spores are dispersed and suspended in a carrier fluid. Is this carrier fluid air or something else? If the spores have to be dispersed, this means that they are collected prior to being used in the chip? So how are they collected?

Answer: Spores are first mixed with water and put into the aerosol generator, which produces the gas mixture of spores and enters the microfluidic chip. It does not mean that spores need to be mixed with the carrier fluid to enter the microfluidic chip. The spores used in the experiment are cultured, not directly obtained from the air, so they must be mixed with water (carrier fluid) first, then into the aerosol generator to produce mixed gas, and finally pumped into the microfluidic chip by the air pump.

13. Line 150: "N-S equation"- Is this the Navier-Stokes equation?

Answer: Yes. The description of this part has been deleted.

14. Line 151: Electroosmotic flow is the flow resulting from the movement of ions in a liquid submitted to an electric potential. It does not apply for air flows, so why take this example? This is confusing.

Answer: This part is written to introduce the Lagrangian method below. In order to avoid misunderstanding, this part has been deleted.

15. Line 160: "according to simulation experiments...". What are "simulation experiments"? Is it computer simulations or experimental measurements? If the authors performed simulations, where are they described? What program was used? What are the conditions?

Answer: The simulation and experiment parts of this paper have been described separately. The simulation in this paper is done with COMSOL Multiphysics 5.1. The design of simulation parameters and the simulation process have been supplemented in 2.2 and 3.1.1.

Lines 163 to 200: The text is impossible to read, because there are so many numerical values in each sentence that it does just not make any sense any more. Please explain the phenomena occurring or the graphs without using numerical values in your sentences.

Answer: A large number of repetitive descriptions of chip size parameters have been deleted. Dimension information description has been placed in Section 2.2 of this article. The readability of articles has been improved.

16. Figure 4 and 5: These graphs are based on measuring the behaviour of 20 particles. This is not much. Please do these experiments with a large enough number of particles (thousands) such that you have valid statistics.

Answer: According to the observation of spore concentration in the literature “Development of a qPCR detection method for monitoring conidial density of rice blast fungus in the air” (DOI: 10.3969/j.issn.1004-1524.2016.08.14), the spore concentration in the air is usually low. When a large-scale outbreak of rice blast spores occurs, the concentration will reach hundreds per cubic meter. The microfluidic chip designed in this paper has an inlet flow rate of 115 ml/min and can extract about 0.17 cubic meters of air in a day. Even in the period of spore outbreak, it can only collect dozens of spores a day, which cannot reach thousands. At this time, spraying control of rice is needed, so hundreds of particles need not be added to the simulation.

17. Line 261 to 268: The authors approximate the spores by oval shapes to calculate the fingerprint of the diffraction pattern. Previously, line143, they explain that they measured the diffraction patterns of 2000 spores to select the threshold levels of the fringes in the corresponding diffraction patterns. This just makes no sense at all. If you have measured 2000 diffraction patterns corresponding to actual spores, why not use them directly? There is no need to approximate the spores by oval shapes.

Answer: When studying the characteristic parameters of the spore diffraction image, the simulation is to study which morphological changes of the diffraction image will be caused by the change of the shape of the target, and to provide a direction for the feature extraction of the spore diffraction image after that. The spore of rice blast is not a circle but a slender shape. It is approximated as an ellipse. The special changes of the diffraction image of the ellipse relative to the circle are studied to find the relationship between the spore morphology of rice blast and its diffraction image morphology. So this simulation process cannot be missing.

18. Figure 276, legend of figure 8: It is not the profile of spore diffraction fingerprint that is shown here but the profile of oval shapes diffraction fingerprint.

Answer: The original figure 8 has been transformed into figure 9. This is the simulation result. The spore diffraction pattern is shown in figure 10.

19. Line 284: The detection accuracy without microfluidic chip is 80%, which is really good. Using the microfluidic chip changes the accuracy to 94%, which is not much more. So this seems to indicate that there is no real need for a microfluidic system at all!

Answer: The detection accuracy of samples without microfluidic chip enrichment is about 80%, and that of samples with microfluidic chip enrichment is 94%. Because of the high concentration of spores in the disposed spore solution and less other impurities, better results can be obtained without using microfluidic chip for enrichment and separation. However, in the field environment, there are many impurities, low spore concentration, and excessive impurities will make the detection accuracy less than 50%. So it is necessary to use microfluidic chip to pre-filter samples before detection. Additional explanations have been made in section 3.2.2 of the article.

20. Lines 287 to 290 confronted to figure 9g: The authors measured 20 samples of spores mixed with pollen and dust, and they seem happy with the results which show a correlation of 0.9912. However, they do not show the results of these experiments in figure 9g, which would have been logical, but the result of another experiment where spores were mixed to beads. Why? Do the authors have something to hide?

Answer: In addition to spores and microparticles, the substances in the experimental samples are more complex. Among them, 2um particles are standard samples with definite parameters of PCR and PVR. The shape and size of rice blast spores were also stable, so the parameters of PCR and PVR were also stable. Pollen and dust varieties are different in size and shape, so they cannot be characterized by the parameters of PCR and PVR, so they are not included in the statistics and only used as interfering substances in the samples.

Reviewer 3 Report

There are few aspects that could be improved in the paper.

The paper needs to be carefully edited for the language. However, some technical aspects are related to the research and those must be addressed before.

The rationale is rather narrow and a more general statement would help t make a case to this proposed solution. His would help the case of the paper. Is only rice that the design could be used for only? May it be extended to other types of grains?

The benefit of performing spectroscopy without lens is not detailed in the introduction as part of the rationale. I agree that PCR is a lengthy process and the proposed method is an improvement to the present approach.

The most important concern I have with the presented paper is the lack of details of how the micro fluidic was conceived. As there is no calculation of any type, I assume that the geometry was based on some previous observations or on trial and error. It would be good to detail this step as the configuration of the microfluidic is an important part of the paper. I understood from the reading that some size analysis was carried out using experimental models. Some details on the choice of size or trend induced by size would be useful in here.

Figure 1 is not self revealing – dimensions should be indicated on the figure.

Separation is mentioned in many places but it is not clear to me after reading the paper about the type of separation or flow in the micro fluidic circuit. Is it pressure flow? Is it electro kinetic flow?

Does enrichment mean collection of more particles joined during flow?

I am afraid that I did not grasp the terminology used in the paper as is a bit non-conventional.

He subject is very interesting and of a great concern but the written paper does not reflect that.

Author Response

There are few aspects that could be improved in the paper.

The paper needs to be carefully edited for the language. However, some technical aspects are related to the research and those must be addressed before.

1. The rationale is rather narrow and a more general statement would help t make a case to this proposed solution. His would help the case of the paper. Is only rice that the design could be used for only? May it be extended to other types of grains?

AnswerThe method studied in this paper can be used not only for spore detection of rice diseases, but also for spore detection of other crop diseases and detection of air pollutants such as PM2.5.

2. The benefit of performing spectroscopy without lens is not detailed in the introduction as part of the rationale. I agree that PCR is a lengthy process and the proposed method is an improvement to the present approach.

AnswerThe introduction has been added to illustrate the advantages of lensless imaging over conventional optical microscopy imaging. This article is indeed an improvement on the previous method. Traditional optical microscopy has complex lens structure, so it is not suitable for large-scale popularization because of its large volume and high price. Its precise optical part is sensitive to external conditions such as temperature and humidity, and is difficult to adapt to complex field environment. Compared with the microscopic imaging mode, lensless imaging technology only uses a complementary metal oxide semiconductor (CMOS) module, which is small in size, stable in performance and not susceptible to environmental impact. Traditional microscopic imaging has a very small field of view (FOV), usually less than 0.5 mm2. Without moving the lens, it is difficult to cover the whole enrichment area of microfluidic chip. The active imaging area of the CMOS sensor is about 20 mm2, which means the FOV is about 20 mm2 and it is approximately 100 times larger than that of a conventional optical microscope.

3. The most important concern I have with the presented paper is the lack of details of how the micro fluidic was conceived. As there is no calculation of any type, I assume that the geometry was based on some previous observations or on trial and error. It would be good to detail this step as the configuration of the microfluidic is an important part of the paper. I understood from the reading that some size analysis was carried out using experimental models. Some details on the choice of size or trend induced by size would be useful in here.

Answer: The principle of separation and enrichment of the first structure is first analyzed. The mixed gas enters the microchannel from the inlet of the microfluidic chip and obtains a horizontal to right initial velocity. If there is no Arc channel 1, the particles cannot get enough centrifugal force to enter the collection tank 1 at a right angle turn. When the Arc channel 1 is added, the Arc channel 1 has a coupling effect with the right angle turn, which increases the centrifugal force of the particles, so that it can enter the collection tank 1. In the simulation, the density of all the particles is the same, so the larger the diameter of the particles, the greater the centrifugal force. The 16um particles can enter the collection tank 1 due to their large centrifugal force, while the 2um particles can enter the second structure. Then the principle of separation and enrichment of the second structure is analyzed. Similar to the first structure, the Arc channel 2 and the right angle turn produce a coupling effect, which increases the centrifugal force of the particles and facilitates entry into the collection tank 2. Reducing L3 accelerates the particles, increasing the acceleration of the 2um particles, and increasing the centrifugal force to enter the collection tank 2. Related content has been added in the paper 3.1.1

4. Figure 1 is not self revealing – dimensions should be indicated on the figure.

Answer: The space in the picture is too small, if the scales are crowded together. Dimension information has been placed in Section 2.2 of this article.

5. Separation is mentioned in many places but it is not clear to me after reading the paper about the type of separation or flow in the micro fluidic circuit. Is it pressure flow? Is it electro kinetic flow?

Answer: In this paper, pressure flow is used to separate and enrich particles.

6. Does enrichment mean collection of more particles joined during flow?

Answer: Enrichment refers to the collection of certain particles into a designated area for subsequent detection.

Round 2

Reviewer 1 Report

The revised manuscript has largely addressed the questions previously raised by the reviewers. However, there are still a few questions as follows:

(1)    In section 3.1.2, the authors mentioned that the smaller particles in collection area 1 were water droplets. From the SEM images in Figure 7 (btw, the authors need to point out the images are SEM images and provide the related imaging experiment procedure in Section 2), the droplets diameter would be about a few microns. Such small water droplets would evaporate and be unstable after the separation experiment. How could the authors still be able to observe the droplets afterwards? Is there any other evidence support the claim of water droplets?
The experimental details of the aerosol generator to produce the sample gas mixture should be provided.

(2)    From the separation principle, I expect the length of channel 2 would be quite important to the separation results. I hope the authors can provide some discussion on this parameter, and the rationale behind the choice of the value of the length of channel 2.

(3)    There are still many spelling and grammar mistakes that need to be taken care of. “um” is still found everywhere in the manuscript including some figures.
